# Elective Surgery in Adult Patients with Excess Weight: Can Preoperative Dietary Interventions Improve Surgical Outcomes? A Systematic Review

**DOI:** 10.3390/nu13113775

**Published:** 2021-10-25

**Authors:** Sally B. Griffin, Michelle A. Palmer, Esben Strodl, Rainbow Lai, Matthew J. Burstow, Lynda J. Ross

**Affiliations:** 1Department of Nutrition & Dietetics, Logan Hospital, Meadowbrook, QLD 4131, Australia; michelle.palmer@health.qld.gov.au (M.A.P.); rainbow.lai@health.qld.gov.au (R.L.); 2School of Exercise and Nutrition Sciences, Queensland University of Technology, Brisbane, QLD 4059, Australia; l20.ross@qut.edu.au; 3School of Allied Health Sciences, Griffith University, Gold Coast, QLD 4215, Australia; 4School of Psychology and Counselling, Queensland University of Technology, Brisbane, QLD 4059, Australia; e.strodl@qut.edu.au; 5Division of Surgery, Logan Hospital, Meadowbrook, QLD 4131, Australia; matthew.burstow@health.qld.gov.au; 6School of Medicine, Griffith University, Gold Coast, QLD 4215, Australia

**Keywords:** surgery, preoperative, complications, obesity, weight loss, VLCD, VLED

## Abstract

This systematic review summarises the literature regarding the impact of preoperative dietary interventions on non-bariatric surgery outcomes for patients with excess weight/obesity, a known risk factor for poor surgical outcomes. Four electronic databases were searched for non-bariatric surgery studies that evaluated the surgical outcomes of a preoperative diet that focused on weight/fat loss or improvement of liver steatosis. Meta-analysis was unfeasible due to the extreme heterogeneity of variables. Fourteen studies, including five randomised controlled trials, were selected. Laparoscopic cholecystectomy, hernia repair, and liver resection were most studied. Diet-induced weight loss ranged from 1.4 kg to 25 kg. Preoperative very low calorie diet (≤800 kcal) or low calorie diet (≤900 kcal) for one to three weeks resulted in: reduction in blood loss for two liver resection and one gastrectomy study (−27 to −411 mL, *p* < 0.05), and for laparoscopic cholecystectomy, reduction of six minutes in operating time (*p* < 0.05) and reduced difficulty of aspects of procedure (*p* < 0.05). There was no difference in length of stay (*n* = 7 studies). Preoperative ≤ 900 kcal diets for one to three weeks could improve surgical outcomes for laparoscopic cholecystectomy, liver resection, and gastrectomy. Multiple randomised controlled trials with common surgical outcomes are required to establish impact on other surgeries.

## 1. Introduction

Obesity has tripled worldwide since 1975, with 31% of adults in Australia classified as obese [1]. The increase in obesity and the associated higher volume of visceral fat and liver steatosis is paralleled by the increased number of patients who require elective surgery for conditions that these individuals are at higher risk for, such as gallstones, liver cancer, and colorectal cancer [2,3].

It is widely accepted that the presence of obesity and excess visceral fat is a risk factor for complications arising from a variety of elective surgeries [4,5,6]. Cardiac and pulmonary events, excess blood loss, surgical site infection, and lengthier stays in hospital occur more frequently for surgical patients with obesity when compared to those in the healthy weight range [5,7,8]. Excess visceral fat and a high liver volume due to liver steatosis are known to complicate the technical aspects of surgery within the abdominal cavity [9]. Furthermore, in minimally invasive surgery, increased abdominal wall adiposity increases the length of time for port placement and hinders abdominal wall compliance during the procedure [6,10]. These difficulties can hinder operative exposure, which can increase blood-loss volume and operating time, both of which are associated with a higher risk of postoperative complications, which attribute major costs to surgical care [11].

Many institutions have implemented formal or informal Body Mass Index (BMI) cut-off thresholds for particular elective surgery types [12,13], and it is now commonplace for surgeons to ask their patients with obesity to lose weight prior to surgery [14]. Despite these requests, there is limited evidence that preoperative weight-loss interventions provide benefit to surgical outcome [15,16]. The only surgical specialty that has been extensively investigated to date is bariatric surgery, where Very Low Calorie Diets (VLCDs) are routinely prescribed to patients for two weeks prior to surgery [10]. Multiple studies in bariatric surgery have demonstrated that preoperative weight loss in response to VLCD, reduces visceral fat and liver steatosis and volume, thereby reducing operating time and risk of some postoperative complications [14]. For other types of elective surgery, it is unclear whether VLCD and/or other types of dietary approaches result in preoperative weight loss and/or liver shrinkage, reduced operating time, and reduced risk of complications. In addition, the type of surgery that may benefit most from risk reduction is presently unknown. Also of concern is the safety of preoperative weight-loss interventions, considering that potential loss of lean body mass may place patients at greater risk of complications [17,18,19].

This systematic review summarises the literature on preoperative dietary interventions for the purpose of weight/fat loss in adult non-bariatric elective surgery patients and their impact on surgical outcomes.

## 2. Materials and Methods

This systematic review followed the Preferred Reporting Items for Systematic Reviews and Meta-Analyses (PRISMA) statement [20] and was registered on PROSPERO (the International Prospective Register of Systematic Reviews) with registration number CRD42021235483.

### 2.1. Search Strategy

A systematic search was performed using electronic databases MEDLINE, Embase, CINAHL, and Cochrane Central Register of Controlled Trials using the search terms defined in Table 1. Clinicaltrials.gov was searched for unpublished trials using basic search terms: “obesity”, “surgery”, and “diet”. Reference lists from eligible articles and known grey literature were manually searched for relevant articles. The literature search was performed in February 2021.

### 2.2. Inclusion and Exclusion Criteria

Articles included in the systematic literature review met all of the following criteria: (i) interventional studies with or without a comparator group; (ii) adults (≥18 years) with overweight/obesity or excess fat (such as centrally distributed fat or fatty liver); (iii) examined any type of elective surgery except bariatric surgery; (iv) evaluated the effect of a preoperative dietary intervention, for the purposes of weight loss, visceral fat reduction, and/or favourable liver changes; and (v) reported on relevant outcomes: weight change, liver changes, intra- or postoperative outcomes or complications, or subjective measures such as surgical difficulty. There were no restrictions on date of publication.

Articles that met any of the following criteria were excluded: (i) not written in English; (ii) animal studies; (iii) used pharmacological or exercise interventions without diet change; (iv) it was unclear if participants had lost weight unintentionally; (v) preoperative weight loss intervention was not described, or it was unclear whether it was dietary in nature; (vi) did not report weight loss or surgical outcomes for participants with overweight/obesity separately; or (vii) outcomes reported were not of relevance to the reviewers (e.g., changes in co-morbidities or laboratory values).

Meta-analysis was planned if an adequate number of studies measured and reported on similar surgical outcomes.

### 2.3. Study Selection

Title and abstract screening for eligible articles was completed independently by the lead author/reviewer (S.G.). Two reviewers (S.G. and R.L.) independently completed full-text screening of all articles against the inclusion and exclusion criteria. Results were compared, and discrepancies between reviewers were discussed with the wider research team to reach a final decision. For studies found via Clinicaltrials.gov search, the author was contacted to request results, if available. 

### 2.4. Data Extraction

Data were extracted independently by S.G. from each included article. Authors of articles were contacted for relevant missing outcome data. Data extracted included characteristics such as author, year of publication, country, setting, and population group studied. Descriptions of dietary intervention and dietitian involvement and measures of adherence and attendance to treatment were also examined. For studies that did not specifically mention presence of a “dietitian”, “dietician”, or “nutritionist” or similar within the intervention, it was assumed there was no dietitian involved. Weight and body measurements or composition change were extracted if reported. Surgical outcomes of interest were extracted where reported and included: mortality, wound complications, and subjective outcomes from the perspective of participants or surgeon(s) such as surgical difficulty. For those studies that had a comparator group, outcomes reported by at least four studies were extracted into a table for ease of comparison and collation; this included: intraoperative blood loss, operating time, infection rate, and length of stay. Data units were standardised for comparison between studies, i.e., length of stay (days), operating time (minutes), infection rates (% incidence), where appropriate.

### 2.5. Quality Assessment

Quality assessment was competed independently by S.G. and R.L. using the “Quality Criteria Checklist: Primary Research” tool created by the American Dietetic Association’s Evidence Analysis Library [21]. Each article was given a quality rating of either “Positive”, indicating issues of bias have clearly been addressed, “Negative”, indicating issues of bias have not been adequately addressed, or “Neutral”, indicating the study had some bias concerns but was neither exceptionally strong nor weak. Results were compared, and the research team consulted on any disagreements to achieve consensus on the final quality rating.

### 2.6. Data Analysis

Meta-analysis was deemed unfeasible due to extreme heterogeneity of intervention type and duration, surgery type, limited number of articles with a comparator group examining comparable outcomes, and missing data despite attempts to contact authors. In its place, where able, a critical synthesis was performed to depict an overview of the size and effects seen for outcomes measured in multiple studies. To create forest plots, a validated calculation by Hozo et al. (2005) was performed to convert values to mean and standard deviation [22] for studies that reported only median and range. Review Manager (RevMan) Version 5.4, The Cochrane Collaboration, London, United Kingdom, 2020, was used to analyse data using pooled mean differences and the inverse-variance method [23].

## 3. Results

### 3.1. Literature Search

Figure 1 outlines the detailed steps of the article selection process as per PRISMA guidelines [20]. From 4597 articles retrieved, 184 duplicates were removed using EndNote X8 (Clarivate Analytics, Philadelphia, PA, USA) and the remaining abstracts screened, with 53 full-text articles assessed for eligibility. A total of 14 studies met inclusion criteria and were included in our systematic review: four randomised controlled trials (RCTs) [24,25,26,27], one pilot RCT [28], five single-cohort [29,30,31,32,33], three retrospective cohort studies [34,35,36], and one retrospective case series [37].

### 3.2. Study Characteristics

Table 2 shows the characteristics of the included studies. Publication dates ranged over 24 years, with the oldest study published in 1997 [33] and the most recent in 2021 [29], with eight of the 14 studies published in the last five years [24,25,26,28,29,30,36,37].

A majority of studies were carried out in westernised countries, with five studies from the United States of America [24,25,31,34,37] and two studies each from Australia [28,29], the United Kingdom [26,30], and Canada [32,35], with one study each from Japan [36], Spain [27], and Finland [33].

The intervention locations were mostly in hospital-based or large medical centre outpatient settings, with most settings described as “hospitals” [26,30,33,36] and some described as surgical specialty facilities [31,37], while some appeared to be university settings [27].

The 14 included studies had a total of 1023 enrolled participants (intervention: *n* = 726, comparator: *n* = 297). Five studies reported dropout rates, which were generally low, the highest being *n* = 4/50 (10%) [28]. The number of participants enrolled in each study ranged from 25 to 188. Participants’ ages ranged from 39 to 71 years of age. Most studies had majority female participants, except for two studies (7% [36] and 45% female [24]).

Eleven out of the 14 studies set inclusion criteria according to BMI cut-offs [25,26,27,28,29,31,32,33,37]. One of these used a combination of BMI and waist circumference [36]. The lowest BMI cut-off was ≥ 25 kg/m^2^ and the highest ≥ 40 kg/m^2^ [24,36]. The remaining studies had eligibility criteria based on the need for hepatobiliary-related surgery with the aim of liver shrinkage or reduction in liver steatosis [33,34,35]. Baseline mean BMI across the studies ranged from 26 kg/m^2^ [36] to 49 kg/m^2^ [31]. The lowest individual BMI was 22.5 kg/m^2^ for a participant who met inclusion criteria with waist circumference ≥85 cm undergoing gastrectomy for gastric cancer and the highest was 85 kg/m^2^, for a complex incisional hernia repair.

General surgery was the most frequently represented surgery type, with nine studies examining a specific general surgery procedure (hernia repair *n* = 3, liver resection *n* = 3, laparoscopic cholecystectomy *n* = 2, gastrectomy *n* = 1). Orthopaedic surgery was limited to knee or hip arthroplasty, examined in two studies [27,37]. The remaining three studies each examined a range of elective procedures. One examined both hernia repairs and laparoscopic cholecystectomies [28], while the other two studies [29,33] each examined a wide range of procedures, including gynaecological, general, orthopaedic, and neurological, but had no control groups. The numbers of participants for these surgeries ranged from six participants in colorectal surgery to 54 participants in gynaecological surgery.

### 3.3. Intervention and Comparator Characteristics

Across the 14 studies, preoperative dietary interventions ranged greatly in duration, method of delivery, and type, but VLCD or a variation of VLCD was most used. See Table 3 for details of interventions. For the 11 studies that reported kcal (kilocalorie) prescription, diets ranged from 458 kcal to 1500 kcal. When kcal of the diets were compared against the accepted definition of a VLCD [38] (≤800 kcal per day), six studies utilised a VLCD [24,26,28,30,31,33], while others used “VLCD-based” (800–1200 kcal) [29] or LCD (900–1190 kcal) [27,34,35]. One study reported using a VLCD, but when examined, only one VLCD meal replacement per day was prescribed, with the remaining two meals of the participants’ choice [36], which does not meet VLCD criteria. Two studies used what appeared to be general healthy eating advice [25,37], and one used a “weight loss program” which was based on a 1500 kcal/day, “low carbohydrate” diet [32]. Surgeries in proximity to the liver utilised VLCD or LCDs exclusively [24,26,30,34,36].

Eight studies utilised comparator groups that most commonly used the “usual intake” of participants (five studies) [24,26,34,35,36]. The remaining three studies implemented a less intensive intervention for comparison: a reduction in usual intake by 500 kcal [27], a “healthy eating information sheet” [28], or “standard counselling without dietary intervention” [25].

Studies reported wide variations in intervention durations, ranging from 1 week [24,34] to 26 weeks [23], with an outlier of 68 weeks [31]. The median duration was 9 weeks when including the outlier. The duration of the dietary intervention differed between participants in five studies due to the need to lose a certain amount of weight (7% [25] or 10% [35] body weight) or reach a certain BMI target prior to surgery (BMI < 40 kg/m^2^ [31,37] or <28 kg/m^2^ [32]). One of these studies aimed to achieve a BMI of <40 kg/m^2^ before complex incisional hernia repair [31], requiring lengthy dieting time frames (68 weeks on food-based VLCD), likely due to the high baseline BMI of participants (49 ± 10 kg/m^2^, range 36–85 kg/m^2^). An additional two studies provided individualised weight-loss targets, and thus duration was based on planned operation, weight-loss progress, and judgement of surgeon [29,33]. All five studies that examined surgery types in proximity to the liver used either VLCD or LCD interventions of less than three weeks [24,26,30,34,36], with the exception of one study, which aimed at weight loss prior to liver donation [35] and used VLCD for a median duration of 7.3 weeks. The two remaining studies had set durations of eight weeks [28] or 12 weeks [27] prior to general surgical procedures and hip/knee replacements, respectively.

Eight studies had a dietitian involved in their interventions: the interventions were led solely by a dietitian (five studies) [24,28,29,36,37]; in the other studies, the dietitian provided dietary advice once at the beginning of the intervention [35], on demand via telephone and a diet sheet [26], and within a wider multidisciplinary team [25]. In three studies, it was unclear who provided dietary instruction in a once-off appointment using verbal advice alone [27,32] or a “diet sheet” [30]. The remaining three studies had instructions delivered solely by their surgeon [34], a “medical weight-loss specialist” [31], or a “therapist” [33]. Interventions of longer than three weeks appeared to incorporate fortnightly or monthly contact with a dietitian or the multidisciplinary team. Dietary protein prescriptions were based on individuals’ body weights in three studies [28,29,31], with other prescriptions ranging from 33 g [34] to 90 g per day [35], and six studies not reporting the amount of protein prescribed [25,26,30,32,36,37].

Only six studies measured adherence to the diet, using a variety of measures from urinary ketone levels [28] to verbal reports from the participant to the surgeon on the day of surgery [34]. Three studies measured adherence via written food records [24,26,27] (ranging from three days to two weeks). Two studies utilised VLCD (≤800 kcal) and found a mean intake of 805 kcal [24] and 947 kcal [26]. The other study utilised the reduction of usual intake by 500 kcal for the intervention group (reduction of 543 kcal achieved) but had similar kcal intake between groups (1290 kcal in comparator versus 1190 kcal in intervention). Three studies included prescribed structured exercise as part of the intervention [25,27,35], and, in one study, structured activity was encouraged throughout [29].

### 3.4. Weight Change and Body Composition

Table 4 outlines the body composition and weight change resulting from the interventions, reported in all except two studies that chose to focus on liver-related changes such as hepatocyte glycogen/steatosis [24] and blood loss [34]. Mean weight loss across the 11 studies that reported mean or median weight change was 8.6 kg (range of 1.4 kg [30] to 24 kg [31]), with a mean of 1.7 kg loss per week over an average of 6.5 weeks for VLCDs (when the outlier of 68 weeks was removed) and 0.4 kg per week for other diets. One study reported BMI loss alone, which was 4.4 kg/m^2^ [35]. Only two studies measured lean body mass change [27,28], for a general surgery cohort and joint replacements, with both finding no statistically significant change from baseline.

### 3.5. Impact on Surgical Outcomes

Table 5 shows the most reported surgical outcomes with a comparator group to allow for analysis. Operating time and blood loss were the only objective surgical outcomes that were found to be impacted significantly. Operating time was reduced in one study by six minutes (*p* < 0.01) for laparoscopic cholecystectomy participants given a VLCD for two weeks preoperatively [26]. This study was the only study out of six that was adequately powered to detect this difference as a primary outcome. Operating time was the only outcome with sufficient data across multiple studies to enable collation into a forest plot (see Appendix A for forest plot). Three of four studies showed reduced blood loss: two liver resection studies that used VLCD or LCD (900 kcal) for two weeks [34] and one gastrectomy study that used a VLCD for three weeks [36]. Blood-loss reduction ranged from −27 mL for gastrectomy [36] to −411 mL for liver resection [24] (*p* < 0.05). Infection rate was measured in four studies, but only two had sufficient data for statistical comparison between groups, and no statistically significant difference in infection rates was found [28,34]. Two other studies reported having no infection occurrences in either group [25,27]. Length of hospital stay was reported in seven studies [24,25,26,27,28,34,35], and none showed statistical significance between groups.

Subjective outcomes were collected in five studies. Four studies surveyed the surgeon regarding the difficulty of the procedure [24,26,29,30]. Due to a lack of validated tools for this purpose, different scales and surveys were used in each study, making it difficult to compare results. Regardless, two of these studies surveyed surgeons performing a laparoscopic cholecystectomy [26,30] who were blinded to groups. The surgeons were asked to evaluate the dissection or visualisation of Calot’s triangle, a critical point in the procedure that can be made technically difficult due to enlarged steatotic left hepatic lobe, excess intra-abdominal fat, and a more friable, fatty liver which is prone to bleeding [40]. In one study, surgeons found visualisation was easier for the intervention group, and in the other study, surgeons found dissection easier in the intervention group (*p* < 0.05 for both studies). One study found a statistically significant increase in health-related quality of life post-intervention from baseline for the intervention group [28].

Eighteen other surgical outcomes were reported on, including mortality, deep vein thrombosis, bile leak, conversion to open procedure, abscess, wound dehiscence, haematoma, and hernia recurrence (for hernia repair), but were infrequently reported across studies. Most had no comparator group, and when able to compare between groups, showed no differences.

### 3.6. Quality Assessment

Figure 2 shows quality assessment outcomes for the included studies against quality criteria. Three studies met all 10 criteria for low-level risk of bias [24,26,28], and six studies received an overall “positive” quality rating. Eight studies received an overall “neutral” quality rating, mostly due to differences between or lack of a comparator group and poor or unclear reporting on whether measurement of outcomes was blinded. Nine studies had at least one criterion considered indicative of a high risk of bias [25,27,30,31,32,33,34,35,36].

## 4. Discussion

This is the first review to summarise the literature on preoperative dietary weight-loss interventions for adult non-bariatric elective surgery patients and their impact on surgical outcomes. Prescription of a preoperative VLCD or low calorie diet (≤900 kcal) that utilised meal replacements for one to three weeks appeared to reduce operating time for laparoscopic cholecystectomy and reduce blood loss for liver resection and gastrectomy. This was supported by the subjective reduced difficulty of dissection and critical view of Calot’s triangle for laparoscopic cholecystectomy. Length of stay did not appear to be affected. These results should be interpreted with caution as no surgery types had multiple RCTs to confirm results. No conclusions could be made regarding the other surgery types examined due to heterogeneity of interventions, surgery types, surgical endpoints, and lack of comparator groups throughout the remaining studies. Unfortunately, there were no studies that specifically examined gynaecological, colorectal, endocrine, urological, or cardiothoracic surgery, which represents a substantial gap in the literature.

It has been established that a 14-day VLCD prior to bariatric surgery can reduce the risk of 30-day postoperative complications [14,41]. The present review highlighted that the same intervention could potentially be utilised prior to elective laparoscopic cholecystectomies, liver resections, and gastrectomy to reduce visceral fat and liver steatosis, leading to reduced blood loss and operating time. Whether it reduces 30-day postoperative complications in the same way as bariatric surgery is still unclear and requires further research. Measuring patients’ visceral fat or liver steatosis is not currently part of standard care for patients awaiting these types of surgery but has the potential to better predict postoperative complications than BMI [42] and may be worth considering to facilitate direction of additional preoperative resources to those patients who would most benefit. 

There have been two systematic reviews published in this area, and both examined bariatric and non-bariatric surgery studies together [15,16]. Due to lack of reporting on common surgical endpoints, they were both unable to draw conclusions on benefits to surgical outcomes, and one review attempted meta-analysis [16], but only three non-bariatric studies were included. In this respect, the present review sheds new light to demonstrate potential risk reduction for three surgical procedures (liver resection, laparoscopic cholecystectomy, and gastrectomy) but also highlights the major gaps in the literature for other surgery types. Given the increasing pressure placed on hospital systems and surgeons to optimise their surgical patients with excess weight due to the rising prevalence of obesity, this research should be prioritised.

Bariatric surgery patients can achieve 2.8 kg to 14.8 kg weight loss with preoperative VLCDs [43]. The present review reflects the same degree of weight loss is possible for non-bariatric surgery patients, with VLCD or VLCD-based intervention achieving 1.4 kg (7-day duration) to 24 kg loss (17-month duration) and at the most rapid rate (1.4 kg/week). The results of the present review parallel conclusions made in numerous studies on VLCD that weight loss appears to correlate with the duration of dietary intervention rather than degree of caloric restriction [15]. The suitability of time-frames and the efficacy of a preoperative weight-loss intervention is important to consider, as delaying elective surgery has been shown to result in deteriorations in health, worsened quality of life, increased disability, and decreased work capability [44,45]. Further, delaying surgeries such as hernia repairs or laparoscopic cholecystectomy can result in unplanned emergency surgery [46,47], which has an inherently higher risk of complications than elective surgery [48]. Due to these risks, it may be prudent to consider a VLCD as the most appropriate dietary choice for adults awaiting surgery for conditions that may worsen over time due to the rapidity of weight loss it provides when compared with other diets.

Few studies in the current review measured common surgical endpoints such as infection or respiratory events, despite the known increased risk of these events for patients with obesity. When measured, very low incidences of these outcomes were found across studies but could not be linked to the preoperative intervention due to underpowered sample sizes and lack of comparative groups. The most commonly reported endpoint in studies with a comparator group was length of stay, with all seven studies showing no significant difference. Despite this, length of stay should continue to be measured in future studies due to the profound implications it has for healthcare facilities and the small sample sizes and number of studies in this review.

Increased BMI is associated with increased costs of surgical care [49,50]. The two outcomes that were found to be impacted by preoperative dietary interventions (blood loss and operating time) have great cost-saving potential. Using operating time as an example, standard operating theatre costs in Australian public hospital facilities, not including equipment costs, are approximately AUD $7 per minute [51]. Therefore, even a seemingly insignificant reduction in operating time per laparoscopic cholecystectomy procedure could have major benefits, with 49,874 laparoscopic cholecystectomy procedures performed in Australia annually [52]. Furthermore, other hospital-acquired postoperative complications can incur punitive financial penalties at some Australian healthcare facilities [53]. A preoperative intervention that potentially reduces operating time and postoperative complications for adults with excess weight could be considered an investment in surgical care, but studies are required to show definitive economic benefit.

There is a clear association between depletion of lean body mass and poor surgical outcomes [54,55,56], including delayed wound healing, infectious complications, intensive care unit admissions, and longer hospital stays [57]. Despite this, there was a distinct lack of reporting on change to lean body mass in the included studies. Although the two studies that measured this reported no excess loss of lean body mass, this result cannot be assumed across all population groups, such as those with malignancy or awaiting organ transplants who may have underlying dysfunctional metabolic processes. Loss of lean body mass can be mitigated by providing adequate dietary protein [18], but interestingly, only three studies individualised their protein prescription based on body weight, and one study provided only 33 g of protein per day, which would be inadequate to meet protein requirements for any adult with obesity when based on minimum 0.8 g/kg. This highlights the need to ensure adequate protein provision in preoperative weight-loss interventions as a matter of urgency, especially considering the increasing pressure placed on patients to lose weight preoperatively [12,13]. Loss of lean body mass could potentially be avoided by utilising qualified dietitians for dietary prescription and monitoring, but due to the lack of studies in this area that involve dietitians and measure lean body mass, this is still unknown and requires exploration.

Study design was a major limitation of the included studies. For example, the lack of comparative data was a source of bias in single-cohort studies [29,30,31,32,33] and retrospective studies [28] and contributed to a lower level of evidence from this review. The non-randomised comparative studies had potential sources of major bias: methods often poorly described with opportunistic recruitment, with historical or unmatched comparator groups. The high number of studies without a comparator group (six studies) and the differing surgery types and length of intervention duration reduced the capacity of the authors to draw conclusions on the impact of their interventions or to facilitate meta-analysis. For future studies, it would seem pertinent to establish a list of relevant surgical endpoints for inclusion to facilitate meta-analysis and definitive conclusions. The American College of Surgeons National Surgical Quality Improvement Program (NSQIP) [58] has developed a list of postoperative complications that are collected and analysed at participating hospitals, which may provide a scaffold for future studies to collect common outcome data.

## 5. Conclusions

Preoperative VLCD and LCD interventions provided for one to three weeks to patients with overweight, obesity, higher amounts of visceral fat, or liver steatosis have potential to reduce operating time and blood loss during non-bariatric elective surgeries within the abdominal cavity, such as laparoscopic cholecystectomy, liver resection, and gastrectomy. Although the benefits were likely in studies that used dietary interventions longer than three weeks in other surgery types, they were less clear due to the lack of comparator groups and common surgical endpoints.

To determine whether preoperative dietary interventions for weight loss improve surgical outcomes, larger well-designed randomised controlled trials need to examine other surgery types and common surgical endpoints. Furthermore, safety of dietary interventions through preservation of lean body mass, particularly for patients with malignancy or underlying metabolic processes, should be confirmed to ensure there is no added risk to patients due to restrictive dieting in the preoperative period.

## Figures and Tables

**Figure 1 nutrients-13-03775-f001:**
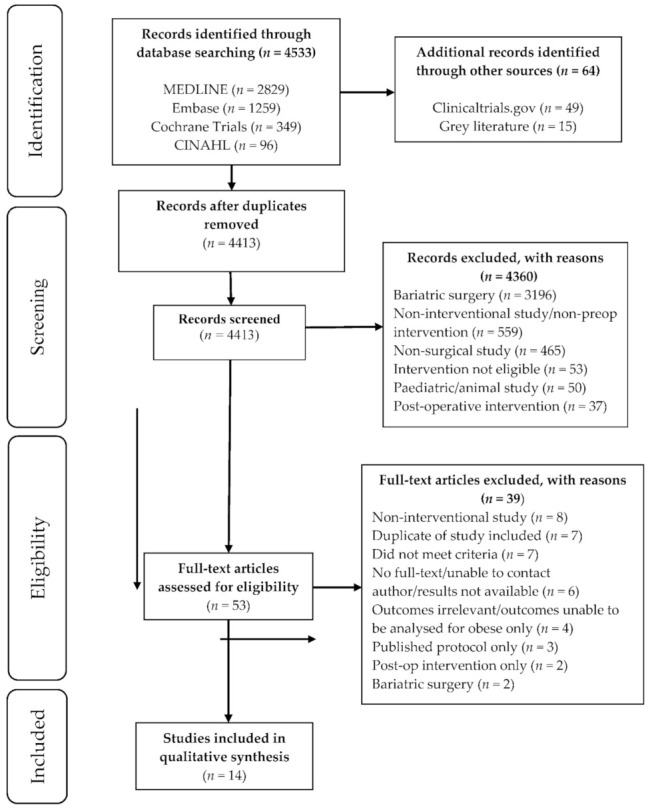
Summary of search strategy and study selection. MEDLINE: a literature database, the primary component of PubMed, developed and maintained by the NLM National Center for Biotechnology Information (NCBI); CINAHL: literature database which indexes the top nursing and allied health literature available.

**Figure 2 nutrients-13-03775-f002:**
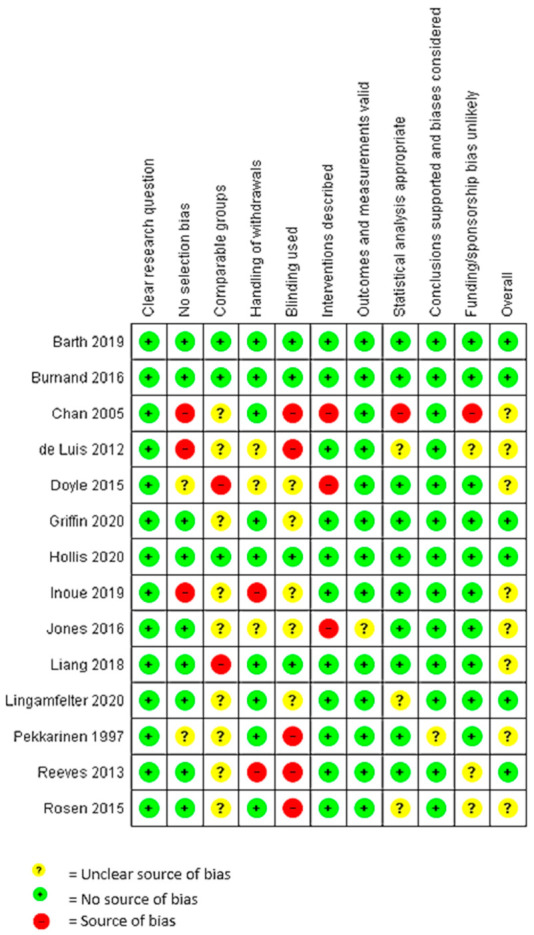
Quality assessment ratings for 14 studies included in systematic literature review.

**Table 1 nutrients-13-03775-t001:** Participants, Interventions, Comparisons, Outcomes, and Study Design (PICOS) criteria.

Parameter	Criteria	Search Terms (Example Taken from MEDLINE Search)
Participants	Adults with obesity or excess weight/fat who are undergoing elective, non-bariatric surgery	obes * or overweight or “body size” or “visceral fat” or “central obesity” or “high body mass index” or adipos * or fat
Interventions	Dietary intervention(s) with/without exercise that were aimed at weight/fat loss prior to surgery	diet or “weight reduc *” or “weight loss” or “preoperative care” or “diet therapy” or “healthy lifestyle”
Comparisons	-	-
Outcomes	Surgical outcome(s) of interest	“postoperative complication *” or “surgery outcome *” or “intraoperative complication *” or “length of operation” or "length of surgery" or “length of stay” or “operat * time” or “intraoperative time” or “surgical procedures, operative” or ischemia or anesthesia or necrosis
Study design	All study types considered	-

* truncation symbol, used at the end of a word to search for all terms that begin with that word root (also called ‘wildcard’).

**Table 2 nutrients-13-03775-t002:** Characteristics of studies assessing effect of preoperative dietary interventions for weight loss or liver changes on surgical outcomes.

Author (Year)	Country	Study Design and Quality	Outpatient Setting	Surgery Type and Procedure	Inclusion Criteria (Aged ≥ 18 Years unless Stated)	Exclusion Criteria	Baseline Demographics, Mean Unless Stated (Range)	Sample Size for Intervention (I) and Comparator (C) Groups, Dropouts
Griffin et al., 2021 [29]	Australia	Single cohort +	Large public hospital	Various-including general, orthopaedic, gynaecological	BMI ≥ 30, referred by surgeon to specialist dietitian clinic	Contraindications to VLCD/medically unsafe/unsuitable for VLCD	90% female Age: 45 ± 13.1 years BMI 44.3 ± 6.2 (33.4–60.4)	I: *n* = 78 No comparator Dropout *n* = 1
Hollis et al., 2020 [28]	Australia	Randomised controlled trial (pilot study) +	Tertiary public hospital	General Surgery-hernia repair or laparoscopic cholecystectomy	BMI ≥ 30 awaiting surgery	Contraindications to VLCD	63% female Age: 51.6 ± 13.1 years BMI 40.5 ± 5.9 (range NR)	I: *n* = 25 C: *n* = 25 *n* = 4 dropouts (*n* = 23 completed in each group)
Lingamfelter et al., 2020 [37]	United States of America	Retrospective case series +	Tertiary specialist institution	Orthopaedic-hip or knee replacement	BMI ≥ 40 requiring surgery	NR	60% female Age: 62.6 ± 8.5 years BMI 44.6 ± 3.9 (40–61.9)	I: *n* = 111 No comparator Dropouts NR
Barth et al., 2019 [24]	United States of America	Randomised controlled trial +	Two large medical centres	Hepatobiliary-liver resection	BMI ≥ 25 requiring surgery	NR	45% female Age: 56 ± 12 years BMI 32.5 ± 6.4 (range NR)	I: *n* = 31 C: *n* = 32 (n *=* 60 analysed, *n* = 3 did not receive surgery)
Inoue et al., 2019 [36]	Japan	Retrospective cohort Ø	Hospital	Upper gastrointestinal-laparoscopic gastrectomy for gastric cancer	Age ≥ 20, BMI ≥ 25 OR waist circumference ≥ 85 cm (men), ≥ 90 cm (women), planned for surgery for stage 1 or 2 cancer	ASA ^β^ score ≥ III, inadequate organ function, hx upper abdominal open surgery, “uncontrolled” diabetes, active infectious disease, on steroids, allergy to shake ingredient(s)	7% female Age: 71 (median) years BMI 26 (median) (23.5–31)	I: *n* = 33 (*n* = 27 had surgical outcomes analysed) C: *n* = 23 Dropouts NR
Liang et al., 2018 [25]	United States of America	Randomised controlled trial Ø	Safety-net academic institution	General surgery-ventral hernia repair	BMI 30–40, hernia 3–20 cm, willing to undergo preoperative optimisation	Severe co-morbidities limiting survival, requiring emergency surgery, pregnant / intending to become pregnant	83% female Age: 49.5 ± 10.1 years BMI 36.8 ± 2.6 (range NR)	I: *n* = 59 (*n* = 3 dropouts, *n* = 12 no surgery), final *n* = 44 C: *n* = 59 (*n* = 1 dropout, *n* = 24 no surgery), final *n* = 34
Burnand et al., 2016 [26]	United Kingdom	Randomised controlled trial +	Hospital	General-laparoscopic cholecystectomy	BMI ≥ 30, symptomatic gallstones	Pre-existing liver disease, diabetes, bile duct stones, previous abdominal surgery	91% female Age: 46 (median) years BMI 33.8 ± 3.4 (range NR)	I: *n* = 21 C: *n* = 25 *n* = 0 dropouts
Jones et al., 2016 [30]	United Kingdom	Single cohort Ø	Hospital	General-laparoscopic cholecystectomy	Any BMI, requiring surgery for biliary colic	Previous cholecystitis or treatment for gallstones, previous upper midline abdominal surgery	87% female Age:45 (median) years BMI 30.3 (median) (22.5–44.1)	I: *n* = 38 (*n* = 19 BMI ≥ 30) No comparator Dropouts NR
Doyle et al., 2015 [35]	Canada	Retrospective cohort Ø	Hospital	Organ transplant-liver	Any BMI, liver donors with >10% liver steatosis (put into intervention group), with <10% steatosis (put into control group)	Suspected non-alcoholic steatohepatitis	58% female Age: 39(mean) years BMI 29.6 (25.4–34.9)	I: *n* = 16 (*n* = 14 had surgery) C: *n* = 53
Rosen et al., 2015 [31]	United States of America	Single cohort Ø	Comprehensive specialty hernia centre	General surgery-complex incisional hernia repair	BMI > 35 who presented to author, evaluated by medical weight-loss specialist	NR	80% female Age: 54 ± 9 years BMI 49 ± 10 (36–85)	I: *n* = 25 No comparator Dropouts NR
Reeves et al., 2013 [34]	United States of America	Retrospective cohort +	Hospital	Hepatobiliary-liver resection	Any BMI, patients who underwent major hepatic resection	NR	42% female Age: 60 (mean) years BMI 27.2 (18.5–47.6)	I: *n* = 51 C: *n* = 60 Dropouts NR
de Luis et al., 2012 [27]	Spain	Randomised controlled trial Ø	Clinical nutrition unit within hospital	Orthopaedic-hip/knee replacement	BMI > 30, indication for surgery for chronic osteoarthritis	Heart disease or stroke in last 3 months, elevated blood lipids or blood pressure > 140/90 mmHg, medications for diabetes, blood pressure, taking steroids	83% female Age: 65 ± 8.5 years BMI 38.6 ± 4.7 (range NR)	I: *n* = 20 C: *n* = 20 (*n* = 42 recruited, *n* = 40 completed, dropout group NR)
Chan and Chan, 2005 [32]	Canada	Single cohort Ø	Hospital specialising in abdominal hernias	General surgery-ventral/incisional hernia repair	Any BMI but BMI > 30 given intervention, undergoing ventral/incisional hernia repair	NR	51% female Age: 56 (mean) years BMI 28.9 ± 13.1 (range NR)	I: *n* = 188 No comparator Dropouts NR
Pekkarinen and Mustajoki, 1997 [33]	Finland	Single cohort Ø	Six metropolitan surgical/ gynaecological hospitals	Mixed, including gynaecological, general, orthopaedic	BMI ≥ 35	Contraindications to VLCD	60% female Age: 50 ± 10.5 years BMI 44 (median) (35–58)	I: *n* = 30 No comparator Dropouts NR

BMI = Body Mass Index, measured in kg/m^2^; ^β^ ASA = Physical Status Classification System to predict perioperative risk. Score I to III; I = healthy; II = mild systemic disease; III = severe systemic disease; IV = severe systemic disease that is a constant threat to life; NR = Not Reported; + = positive quality score [21]; Ø = neutral quality score [21].

**Table 3 nutrients-13-03775-t003:** Dietary intervention and weight-loss outcomes of studies assessing effect of dietary-induced weight loss or liver changes on surgical outcomes.

Author (Year)	Diet Type, Product, Funding	Dietary Profile	Dietary Prescription	Duration	Service Provided	Frequency of Contact and Attendance	ADHERENCE TO DIET	Tolerance/Acceptance	Weight Change for I (Intervention) and C (Comparator) Group (Loss Unless Stated)
Griffin, et al., 2021 [29]	VLCD-based-Optifast^®^/Optislim^®^ shakes, bars, soups, or desserts. Participants paid for own products.	800 kcal–1200 kcal, 0.8–1 g/kg adjusted body weight protein, fat NR.	1–4 meal replacements, >2 cups non-starch vegetables, 2 L energy-free fluids, protein-rich foods to meet requirements.	Median 10 weeks to reach target weight. Duration based on amount of weight loss required or time-frame until surgery.	Diet prescription by dietitian individualised to aid adherence. Evidence-based care provided: managing symptoms, education on progression to healthy eating post-surgery.	Fortnightly dietitian appointments, 93% of appointments attended.	Adherence not formally measured but informed the nutrition care plan.	*n* = 1 could not tolerate products, *n* = 2 preferred food, 56% reported ≥1 mild side effect, 90% resolved by end of treatment.	I: 9 ± 6.4kg 7.4 ± 5.3% BW, BMI = 3.3 kg/m^2^. No comparator.
Hollis et al., 2020 [28]	VLCD using Optifast^®^ shakes. Products free of charge.	700–800 kcal with 0.75 g/kg adjusted body weight protein, fat NR.	VLCD shakes (number dependent on meeting protein requirements) >2 cups non-starch vegetables, 2 L energy-free fluids, 1 tsp oil.	8 weeks.	Dietitian-led. Individualised program with advice on managing symptoms.	Fortnightly dietitian appointments, 93% attendance in intervention group vs. 39% in controls (*p* < 0.001).	Urinary ketones found in 56% of intervention participants.	*n* = 1 drop-out due to “gastrointestinal upset”.	I: 6.5 kg ± 3.8, 5.5%BW. C: gain 0.1 kg ± 2.6.
Lingamfelter et al., 2020 [37]	Healthy eating advice.	NR	NR	Mean duration 22 weeks, based on goals BMI < 40.	Individualized meal plans and recommendations for diet changes led by dietitian.	Monthly dietitian appointments; 83% of participants attended >1 appointment.	NR	NR	I: 5.8 ± 5.3 kg %BW = NR BMI = 2.1 kg/m^2^. No comparator.
Barth et al., 2019 [24]	VLCD using Optifast 800^®^ shakes, liquid-only diet. Products free of charge.	800 kcal, 70 g protein, 100 g CHO, 20 g fat.	Five shakes, unlimited energy-free fluids.	1 week.	Dietitian-led. Provided with food-based equivalent VLCD if products not tolerated.	Initial appointment plus two phone calls during the week to ensure adherence and accurate food recording.	Food diaries analysed by dietitian: 28/30 (94%) adherence (consumed solely Optifast^®^, *n* = 1 had diet-adherent food), mean intake 805 kcal and 70 g protein.	*n* = 1/31 pts never started diet (reason NR).	NR
Inoue et al., 2019 [36]	VLCD using Obecure^®^ shake (one per day). Product free of charge.	NR—only product profile reported (178 kcal, 22 g protein, 15 g CHO, 2 g fat per product).	One meal replacement to replace “main meal”, no restriction on other two meals. Low-calorie vegetables with the replaced meal were allowed.	3 weeks (20 days).	Dietitian-led. Dietitian provided “nutritional counselling” for 7 days, then VLCD for 20 days with information on a “nutritionally-balanced” diet, low-calorie foods, and appropriate mealtimes.	Unclear—at least twice during the one month preoperative period.	Adherence reported to dietitian and surgeon = 96.9%.	*n* = 1 stopped VLCD due to taste “intolerable”, no adverse events associated with VLCD.	I: 3.2 kg 4.2% BW BMI = 1.2 kg/m^2^. C: NR.
Liang et al., 2018 [25]	Healthy eating and exercise-“prehabilitation”.	NR	Dietary modifications, daily goals checklist (servings of fruit and vegetables).	Up to 26 weeks, based on body-weight-loss goal of 7%.	Multidisciplinary team: surgical specialists, medical weight-loss experts, dietitians, physical therapists, health educators, nurse practitioners, and study coordinators.	Weekly group meetings plus monthly assessments.	NR	NR	I: 2.72 ± 5.3 kg C: 2 ± 3.8 kg (Non-significant between groups, *p* = 0.308).
Burnand et al., 2016 [26]	VLCD using SlimFast^®^ shakes. Funding source NR.	800 kcal. Protein, CHO, and fat NR.	Two shakes, one ready-made meal of <3% fat (participants’ choice-not described).	2 weeks.	Diet sheet given with diet instructions. Dietitian available to participants via phone.	NR	Via detailed dietary survey, 2-week duration-mean intake 947 kcal/day.	VLCD “well tolerated”. Limited detail.	I: 3.48 ± 1.98 kg BMI = 1.29 ± 0.74 kg/m^2^. 3.8%BW C: 0.98 ± 1.67 kg BMI= 0.36 ± 0.61 kg/m^2^ 1.1%BW (kg and BMI difference between groups *p* < 0.0001).
Jones et al., 2016 [30]	VLCD, product not named. Funding source NR.	800 kcal, protein/ CHO/fat NR.	NR	2 weeks.	Verbal advice to adhere to VLCD (unclear by whom), diet sheets with dietary suggestions provided.	Once at beginning.	NR	NR	I: Median −1.4 kg (range gain 3 kg to 4.2 kg loss). No comparator.
Doyle et al., 2015 [35]	VLCD using Optifast 900^®^. Funding source NR.	900 kcal, 90 g protein, 67 g CHO, fat NR.	Four shakes, up to 2 L water plus coffee/tea without milk or sugar. No other foods/ drinks allowed.	4 to 15 weeks, median 7.3 weeks, guided by BMI reduction target of 10%.	Dietitian saw participants prior to starting diet. No other details given.	Unclear, seen by dietitian at least once at beginning.	NR	Constipation *n* = 2, no adverse effects, no diet dropouts.	I: BMI = 4.4 kg/m^2^ C: NR.
Rosen et al., 2015 [31]	“Protein sparing modified fast” (food-based VLCD).	800 kcal, 1.2–1.4 g protein/kg ideal body weight, 40 mg (authors question accuracy) CHO, fat NR, vitamin and mineral supplementation.	Food-based using “high biological value” protein.	Mean 68 weeks (range 26–144), guided by patient-directed weight-loss goal aiming BMI < 40.	Care led by medical weight-loss specialist (unclear if nutritionally trained). “Nutritional team” administered the diet.	Goals and progress discussed at 3-monthly appointments with surgeon.	NR	NR	I: 24 ± 21 kg (range 2–80 kg) BMI = 9 ± 8 kg/m^2^ (range 0.6–33 kg/m^2^) 18 ± 12%BW. No comparator.
Reeves et al., 2013 [34]	Low-calorie diet using SlimFast^®^ shake. Funding source NR.	900 kcal, 33 g protein, fat 20–40%, and carbohydrate 30–50% of total daily calories.	Breakfast: 1 cup oatmeal/ SlimFast; Lunch: ¾ cup cottage cheese/SlimFast; Dinner: 85 g lean meat/fish/ SlimFast; Allowed ½ cup fruit per meal.	1 week.	Diet information provided by surgeon.	Seen once by surgeon at beginning.	Adherence measured by surgeon asking patient on day of surgery- ”Nearly all” participants indicated they were completely adherent.	NR	NR
De Luis et al., 2012 [27]	Low-calorie diet using Optisource^®^, formula unclear-“envelope”. Funding source NR.	1190 kcal, 63 g protein, 166 g CHO, 21 g fat.	Lunch and dinner meals replaced with Optisource, no limitations on other meals.	12 weeks.	NR	NR	Dietary intake measured via 3-day food records: mean intake 1248 kcal/day (reduction of 543 kcal from baseline), 70 g protein, 163 g CHO, 35 g fat.	NR	I: 7.6 kg BMI = 3.1 kg/m^2^ C: 4.3 kg BMI= 2.1 kg/m^2^.
Chan and Chan et al., 2005 [32]	“Weight loss program”-low-calorie diet (food-based).	1500 kcal, “low carbohydrate”, protein and fat NR.	Diet “limits carbohydrate” and encourages fruit and vegetables.	Median 17 weeks (range 1–530 weeks), guided by weight-loss goals of BMI < 28.	Weight-loss target and diet were explained at initial office visit, unclear by whom.	Once at beginning.	NR	NR	I: 11.5 ± 3.5 kg. 63% of obese reduced their BMI to <30 kg/m^2^. No comparator.
Pekkarinen and Mustajoki et al., 1997 [33]	VLCD using Modifast Sandoz^®^ shake. Funding source NR.	458 kcal, 52 g protein, 45 g CHO, 7 g fat.	3 shakes, small amount of low-CHO vegetables, 2 L water/low-energy fluids. “Re-feeding” period commenced 1 month prior to surgery—reduced to 1 shake per day for 1 week, then to normal food.	Mean 14 weeks (range 7–24), guided by initial weight, planned operation, and weight-loss progress.	“Therapist” provided care, tailored appointments, including “nutritional education and behaviour therapy to create new eating habits and to prevent weight gain”.	Fortnightly.	Listed for those who experienced adverse outcomes/problems with adherence. *n* = 1 depression, *n* = 3 excess alcohol intake. *n* = 6 “suspected” intake of normal food.	*n* = 2 ceased VLCD within “first weeks”, reason NR.	I: 19.6 kg (range 4.3–44.7 kg) BMI = 7 kg/m^2^ 15%BW. No comparator.

BMI = Body Mass Index, in kilograms divided by metres (height), squared; BW = Body Weight; CHO = Carbohydrate; kcal = kilocalories; NR = Not Reported; VLCD = Very Low Calorie Diet.

**Table 4 nutrients-13-03775-t004:** Body composition and surgical outcomes resulting from preoperative dietary interventions for weight loss or liver changes.

Author (Year), Surgery Performed	Intervention (I) and Comparator (C) Group Diets	Body Measurement/Composition Changes	Mortality	Wound/Other Complications	Subjective/Qualitative Outcomes
Griffin et al., 2021 [29] Various procedures including gynaecological, general, and orthopaedic	I: VLCD-based (800–1200 kcal). No comparator.	NR	NR	NR	Surgeon survey: changes from VLCD (*n* = 12): Facilitated access to organ(s): 75% agreement Shortened operating time: 75% agreement Reduced blood loss: 58% agreement Easier laparoscopic access: 58% agreement
Hollis et al., 2020 [28] Hernia repair/laparoscopic cholecystectomy	I: VLCD (800–900 kcal). C: Generic healthy eating information sheet.	I: WC = 6.1 ± 4.8 cm loss MM = 1.53 kg loss FM: 4.48 kg loss C: WC = 1.3 cm gain MM = 0.1 kg loss FM = 0.16 kg loss Both MM and FM: sig. difference between groups (*p* < 0.05)	(Deaths at 30 days) Both groups: 0%	NR	Health-related quality of life: I: +18 from baseline (*p* < 0.05) (improved) C: −1.8 from baseline (NS)
Lingamfelter et al., 2020 [37] Total hip/knee arthroplasty	I: Healthy eating. No comparator.	NR	NR	Delayed wound healing *n* = 2 (2.7%) Superficial infection: *n* = 1 (1.4%) Haematoma *n* = 1 (1.4%)	NR
Barth et al., 2019 [24] Partial hepatectomy	I: VLCD using Optifast 800^®^ (800 kcal). C: Usual intake.	Hepatocyte glycogen score ^β^ = 1.6 in diet group vs. 2.5 in comparators (*p* < 0.0001). No significant difference in steatosis/steatohepatitis between groups.	(Deaths at 30 days) Both groups: 0%	Abscess: 3% of entire cohort Wound dehiscence: 6% of entire cohort	Surgeon blinded to groups: Judged the liver to be easier to mobilise and manipulate in the diet group: 1.86 versus 2.90 (*p* < 0.004)
Inoue et al., 2019 [36] Gastric resection for gastric cancer	I: VLCD using Obecure^®^ (one/day). C: Usual intake (historical controls).	I: WC = −2.74 cm Waist:hip ratio reduced 0.01 cm (both reduced from baseline (*p* < 0.05) C: NR	I: “Hospital mortality” 0% C: NR but reported as NS between groups	Complications only reported for intervention: Pancreatic fistula (12.1%) Anastomotic leak/haemorrhage *n* = 2 (3%) Gastroparesis/Hepatobiliary disorder *n* = 2 (3%)	NR
Liang et al., 2018 [25] Ventral hernia repair	I: Healthy eating and exercise. C: “Standard counselling” without specific dietary change.	I: WC 4.6 ± 16.7 cm loss Hip circumference: 2.1 ± 6.5 cm loss C: WC 1.6± 8.9 cm Hip circumference: 2.3 ±8.4 cm loss Between groups: Waist: NS, *p* = 0.188 Hip: NS, *p* = 0.239	NR	Haematoma I: *n* = 1/44 (2%) C: *n* = 0. Wound dehiscence I: *n* = 1 (2%) C: *n* = 0. Seroma I: *n* = 1 (2%) C: *n* = 0	NR
Burnand et al., 2016 [26] Laparoscopic cholecystectomy	I: VLCD using SlimFast^®^ (800 kcal). C: Usual intake.	NR	NR	Haematoma *n* = 1 in intervention group Not analysed for significance	Surgeon blinded to groups judged the difficulty of procedure-scale 1 (very difficult) to 4 (easy): Dissection of Calot’s triangle easier in intervention group (*p* < 0.02) No difference for other measures: ease of gallbladder retraction, ease of liver displacement, and liver friability
Jones et al., 2016 [30] Laparoscopic cholecystectomy	I: VLCD (800 kcal). No comparator.	NR	NR	NR	Surgeon (blinded to dietary compliance-“compliant” = loss of >2 kg): Compliant participants had better (easier) scores for all areas of difficulty (*p* = 0.018) Difficulty of visualization of Calot’s triangle and gallbladder bed dissection was higher in non-compliant participants but was NS (*p* = 0.07) All cases where the surgeon was unable to retract the liver > 90° (15%, *n* = 6/38) were non-compliant No difficulties were reported in retracting the liver or visualising Calot’s triangle in compliant participants vs. difficulties reported for 42% of non-compliant participants. (*p* < 0.05)
Doyle et al., 2015 [35] Liver donation (recipients)	I: VLCD using Optifast 900^®^ (900 kcal). C: Usual intake.	NR	(Deaths at 90 days) I: *n* = 1 (8%) C: *n* = 1 (2%) (NS difference)	Wound complications: I: *n* = 1(8%) C: *n* = 1 (2%) Abdominal bleed/infection/ bile leak: I: *n* = 6 (46%) C: *n* = 13 (37%) Not analysed for significance separately. Aggregate of all complications = *p* = 0.11 (NS)	NR
Rosen et al., 2015 [31] Complex incisional hernia repair	I: Food-based VLCD (<800 kcal). No comparator.	NR	NR	Seroma *n* = 1 (4%) Surgical site infection *n* = 1 (4%) Deep infection *n* = 1 (4%)	NR
Reeves et al., 2013 [34] Liver resection for liver cancer	I: LCD using SlimFast^®^ and/or food (900 kcal). C: Usual intake.	NR	(Deaths at 30 days) *n* = 0 both groups	Complications per Clavien^£^ scale I: *n* = 12 (25%) C: *n* = 15 (25%) Infectious complications I: *n* = 9 (18%) C: *n* = 6 (10%) NS between groups	NR
de Luis et al., 2012 [27] Hip/knee arthroplasty	I: LCD using Optisource^®^ (1190 kcal). C: Reduction of usual intake by 500 kcal (mean 1290 kcal).	I: WC = −5.5 cm Fat Free Mass = −700 g (NS) FM = −5.6 kg C: WC = −4.2 cm Fat Free Mass = −1.2 kg (NS) FM = −4 kg Waist: hip ratio change: both groups NS	(Time-frame NR) *n* = 0 both groups	DVT: *n* = 1 each group (NS)	NR
Chan and Chan et al., 2005 [32] Ventral incisional hernia repair	I: Food-based LCD (1500 kcal). No control group (but non-obese not given diet).	NR	Perioperative deaths *n* = 0 Postoperative deaths NR	Postoperative hernia recurrence: Obese (intervention group): *n* = 5/72 (7%) Non-obese: *n* = 8/116 (6.9%)	NR
Pekkarinen and Mustajoki et al., 1997 [33] Various procedures including orthopaedic, general, and cardiac	I: VLCD using Modifast Sandoz^®^ (459 kcal). No comparator.	NR	NR	Haematoma *n* = 2 (13%) Surgical site infection *n* = 3 (20%)	NR

BW = Body Weight; CI = Confidence Interval; DVT = Deep Vein Thrombosis; IQR = Interquartile Range; kcal = kilocalories; kg = kilograms; LCD = Low-Calorie Diet; mL = millilitres; MM = Muscle Mass; NR = Not Reported; NS = Non-Significant *p* value; PE = Pulmonary Embolism; VLCD = Very Low Calorie Diet; WC = Waist Circumference; £Clavien scale = deviations from a normal postoperative recovery [39]; ^β^ Hepatocyte glycogen = storage of carbohydrate, lost with ketosis. One gram of glycogen binds to four grams of water resulting in significant loss of liver volume when glycogen stores reduced.

**Table 5 nutrients-13-03775-t005:** Difference in surgical outcomes for patients receiving preoperative dietary intervention with comparator groups.

			Operating Time (mins)			Blood Loss (mL)			Infection Rate (%)			Length of Stay (Days)		
Author, Diet Used, Surgery Type	I n	C n	I	C	Difference for I Group ^€^	I	C	Difference for I Group ^€^	I	C	Difference for I Group ^€^	I	C	Difference for I Group ^€^
Hollis et al., 2020 [28], VLCD, Mixed General	20	14	89.9 ± 29.3	107.5 ± 41.4	−17.6 min NS, *p* = 0.16 ^a^	NR			0	14.2	−14.2% NS, *p* = 0.16 ^b^	1.2 ± 0.5	1.4 ± 0.7	−0.2 days NS, *p* = 0.47 ^a^
BarthBarth et al., 2019 [24], VLCD, Liver resection	30	30	246 ± 72	258 ± 90	−12 min NS, *p* = 0.42 ^c^	452	863	−411 mL (***p* < 0.05**) ^c^	NR			5 (median)	4 (median)	+1 day NS ^c^
Inoue et al., 2019 [36] VLCD, Gastrectomy	27	23	355 (196–567) (median, range)	374 (258–482) (median, range)	−19 min NS, *p* = 0.35 ^d^	49 (1–282) (median, range)	76 (34–914) (median, range)	−27 mL (***p* < 0.05**) ^d^	6.1	NR	NR	NR		
Liang et al., 2018 [25], Healthy Eating, Ventral hernia repair	44	34	102.3 ± 55.8	91.0 ± 49.2	+11.3 min NS, *p* = 0.414 ^c^	NR			0	0	nil	0 (median) (0.2 IQR)	0 (median) (0.1 IQR)	No difference NS, *p* = 0.687 ^e^
Burnand et al., 2016 [26] VLCD Laparoscopic cholecystectomy	21	25	25 (18–41) (median, range)	31 (20–170) (median, range)	−6 min *(**p*** **< 0.01**) ^f^	NR			NR			0.4 (0.2–6.1) (median, range)	0.3 (0.2–1.2) (median, range)	−0.1 day NS, *p* = 0.36 ^f^
Doyle et al., 2015 [35], VLCD Liver resection for donation	14	53	NR			417 (range 255–579)	358 (range 286–429)	+59 mL NS ^g^	NR			16 (mean) (7.8–24.7 95% CI)	18 (mean) (11.4–23.9 95% CI)	−2 days NS, *p* > 0.99 ^g^
Reeves et al., 2013 [34], LCD, liver resection	51	60	NR			600 (51), mean, SEM	906 (76), mean, SEM	−306 mL (***p* < 0.05**) ^h^	18%	10%	−8% NS ^h^	6.3 (1.1), mean, SEM	6.3(1.2), mean, SEM	No difference NS ^h^
De Luis et al., 2012 [27], LCD, Hip/knee replacement	20	20	86.5 ± 21.8	91.8 ± 41.7	−5.3 min NS ^i^	NR			0	0	nil	8.1 ± 2.7	8.9 ± 6.7	−0.8 days NS ^i^

CI = Confidence Interval; I = Intervention Group; IQR = Interquartile Range; LCD = Low-Calorie Diet; n = number of participants; NR = Not Reported sufficiently for analysis; NS = Non-significant; SEM = Standard Error of the Mean; VLCD = Very Low Calorie Diet. Statistical significance set at *p* < 0.05 for all studies; ^€^ when compared to the comparator group; ^a^ independent *t*-test or Mann–Whitney U test as appropriate; ^b^ chi-square or Fisher’s exact test were used as appropriate; ^c^
*t*-test used; ^d^ Wilcoxon test used; ^e^ Wilcoxon rank-sum test used; ^f^ Mann–Whitney U test used; ^g^ independent sample t-test used; ^h^ Student’s *t*-test and Wilcoxon rank-sum test were used as appropriate; ^i^ two-tailed, paired Student’s *t*-test used.

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
