# Peer review of "Elective Surgery in Adult Patients with Excess Weight: Can Preoperative Dietary Interventions Improve Surgical Outcomes? A Systematic Review"

_nutrients, 2021, doi:10.3390/nu13113775_

Round 1

Reviewer 1 Report

The authors should be congratulated for their very well conducted systematic review on preoperative dietary intervention before surgery. This systematic review focused on adults with obesity/excess weight who were undergoing elective non-bariatric surgery with preoperative dietary intervention and describing the surgical outcome.

I would just have a few overall comments:

  • The litterature and results presented are mainly including abdominal surgery and the data on other surgery such as orthopedics are sparse. Would it be better to gain further in clarity to include only abdominal surgery?
  • The main of the systematic review was to analyse surgical outcome. The results presented are principally on perioperative time, perioperative blood loss, and perioperative assessement of difficulty. However, can these data really be considered as surgical "outcomes", or are there more perioperative measurements?

Minor comments:

  • Table 3: Griffin "2021" instead of "2020"
  • Supp Fig 1: The title and legend should be more detailed in order to fully understand the forest plot, including the area concerned (type of surgery,...).

Author Response

The authors thank Reviewer 1 for their valuable comments.

In regard to limiting the review to abdominal surgery (point one)

The authors feel that due to the scarcity of studies on non-bariatric surgery which examine outcomes resulting from preoperative diet-induced weight/fat loss, that omitting surgeries outside of the abdominal discipline for this review would not be desirable. There is robust literature regarding the higher risk of complications from joint replacements in patients with obesity, and as a result many institutions and surgeons have a BMI cut-off placed on these types of surgery. As there is a lack of consensus around best approach to weight loss, the authors thought it important to include all elective, non-bartiatric surgery types in the literature search. Restricting the surgery type to abdominal may also limit the readership of this article to general surgeons only, which is undesirable considering the known growing pressure and demand across other surgery types such as gynaecology, urology and orthopaedic for patients with obesity to lose weight preoperatively. We hope this response satisfies the reviewer, and we have decided to keep all surgery types in this review.

Regarding analysing surgical outcomes (point two):

The authors have attempted to make it clear within the text of this review that there was insufficient data for many surgical outcomes across multiple studies to determine any consistent impact, including mortality, infection, and wound complications. These outcomes have been included in Table 4 and have been compiled together into one column ‘Wound/other complications’. These were the only other relevant outcomes available to report, and per the eligibility criteria for articles, only articles with relevant surgical outcomes were included. We thank the Reviewer for this observation and agree that Table 5 focuses on the outcomes they describe in their comment. These outcomes were the only ones which were measured across four studies, as outlined on Line 125: ‘For those studies which had a comparator group, outcomes reported by at least four studies were extracted into a table for ease of comparison and collation’. The authors believe that unless infection rates etc were compared against usual infection rates for that country or setting, studies which did not have a comparator should not be used to evaluate the impact of the intervention on surgical outcomes.

Minor comments:
Table 3 has now been changed to read ‘2021’ instead of 2020 as requested.

Supp Figure 1 – the title and legend has now been edited for more detailed information to fully understand the forest plot, as requested. I have uploaded the new file attached to this Reply

Reviewer 2 Report

The objective of the review is relevant highlighting the importance of preoperative dietary interventions to improve surgical outcomes

The work is well planned, the objective of the review is of relevant interest

Author Response

Response to Reviewer 2:

The authors thank Reviewer 2 for their feedback.